# The Relationship between Leisure-Time Sedentary Behaviors and Metabolic Risks in Middle-Aged Chinese Women

**DOI:** 10.3390/ijerph17197171

**Published:** 2020-09-30

**Authors:** Jing Fan, Caicui Ding, Weiyan Gong, Fan Yuan, Yanning Ma, Ganyu Feng, Chao Song, Ailing Liu

**Affiliations:** Department of Nutrition and Health Education, National Institute for Nutrition and Health, Chinese Center for Disease Control and Prevention, Beijing 100050, China; jing_zwtcheroyl@163.com (J.F.); dingcc@ninh.chinacdc.cn (C.D.); gongwy@ninh.chinacdc.cn (W.G.); yuanfan@ninh.chinacdc.cn (F.Y.); mayn@ninh.chinacdc.cn (Y.M.); fenggy@ninh.chinacdc.cn (G.F.); songchao@ninh.chinacdc.cn (C.S.)

**Keywords:** sedentary behaviors, metabolic diseases, women’s health

## Abstract

The prevalence of metabolic diseases has increased over the past few decades, and epidemiological studies suggest that metabolic diseases may be associated with lifestyle. The purpose of the present study was to investigate the relationship between leisure-time sedentary behaviors (LTSBs) and metabolic risks in middle-aged women in China. Data came from the China National Nutrition and Health Surveillance (CNNHS) in 2010–2012. A total of 2643 women aged 46 to 53 years were involved. Multiple linear regression was used to examine the association of leisure-time sedentary duration (LTSD) with total cholesterol (TC), triglyceride (TG), waist circumference (WC), and body mass index (BMI). Restrictive cubic splines (RCS) were used to plot the curves between LTSD and the risk of metabolic diseases. Region, education, income, alcohol consumption, exercise, daily energy intake, and fat energy ratio were adjusted for all models. After adjusting for potential influencing factors, the results of multiple linear regression showed that for each additional hour increase in LTSD, TC and TG increased by 0.03 mmol/L and 0.04 mmol/L, respectively. The results of RCS curves showed that the risks of MetS (*p* for trend = 0.0276), obesity (*p* for trend = 0.0369), hypertension (*p* for trend = 0.0062), and hypercholesteremia (*p* for trend = 0.0033) increased with the increase in LTSD. LTSB was associated with the risks of MetS, obesity, hypertension, and hypercholesteremia in middle-aged women. Reducing LTSD may be an effective way of preventing metabolic diseases in middle-aged women.

## 1. Introduction

Globally, metabolic diseases are gradually increasing in prevalence all around the world. The number of adults with diabetes has increased by about three times since 1980, from 108 million in 1980 to 422 million in 2014 [1]. Each year, diabetes leads to 1.5 million deaths worldwide. The World Health Organization (WHO) estimates that dyslipidemia contributes to one-third of global ischemic heart disease and one-fifth of global cerebrovascular disease, which is equivalent to nearly 2.6 million deaths worldwide each year [2]. From 1990 to 2015, the rate of hypertension increased from 17,307 to 20,526 per 10,000 persons, and the estimated rate of annual deaths related to hypertension increased from 97.9 to 106.3 per 10,000 persons [3]. Meanwhile, hypertension led to a loss of disability-adjusted life years (DALYs) from 95.9 million to 143.0 million around the world [3]. Over the past three decades, the prevalence of metabolic syndrome (MetS) has increased in many countries. It is estimated that about one billion people worldwide are suffering from MetS [4]. The Consortium of Global Burden of Disease Obesity analyzed data on adults in 195 countries between 1980 and 2015, showing that the global prevalence of overweight and obesity continued to increase during those 35 years and about four million deaths were directly related to high body mass index (BMI) [5]. The prevalence of metabolic diseases is also on the rise in China. From 2002 to 2012, the prevalence of overweight and obesity increased from 22.8% and 7.1% to 30.1% and 11.9%, respectively, and the prevalence of diabetes and hypertension rose from 4.2% and 18.8% to 9.7% and 22.8%, respectively [6]. During that decade, the number of patients with MetS increased by about 50 million, and the prevalence of dyslipidemia increased from 18.6% to 40.4% [6]. Between 6% and 30% of chronic diseases in China can be attributed to overweight and obesity, with the direct economic burden as high as CNY 90.768 billion [7]. The DALYs caused by hypertension in the Chinese population has reached 37.94 million years [8]. Diabetes is a major cause of blindness, kidney failure, cardiovascular and cerebrovascular accidents, and amputation in China, and the burden of disease is heavy [9]. It is predicted that elevated total cholesterol (TC) levels will lead to an increase of 9.2 million cardiovascular events in China between 2010 and 2030 [10].

The increasing prevalence of metabolic diseases may be due to changes in lifestyle, including dietary habits, physical activities, and sedentary behavior (SB), where the role of SB for metabolic risks is gradually being recognized. Leisure-time sedentary behaviors (LTSBs) which are a component of SB, refer to sedentary activities in leisure time including reading, watching television, using a computer or smart phone, and other screen-based pastimes [11]. The association of LTSB with metabolic diseases such as type 2 diabetes [12,13], obesity [14,15], dyslipidemia, elevated blood pressure (BP) [16,17], and MetS [18,19] was investigated in previous studies. Independent of physical activity, LTSBs, especially TV viewing, were strongly associated with increased risk of type 2 diabetes and obesity [12]. Hormonal changes in middle-aged women may adversely affect metabolic and cardiovascular processes, such as reduced energy consumption, elevated fasting insulin levels, and elevated high-density lipoprotein cholesterol (HDL-C) levels, making them more susceptible to many metabolic disorders [20,21]. In addition, women at that age are less physically active than men, which is related to the roles they play and workloads they undertake in society and at home [22]. Cardiovascular diseases account for half of all deaths among women over 50 in developing countries, and more than 70 million women worldwide are affected by diabetes [22]. Thus, our study focused on estimating the relationship between leisure-time sedentary duration (LTSD) and a variety of metabolic diseases in middle-aged women.

At present, large sample studies or relatively comprehensive studies on the relationship between SB and metabolic diseases were mainly conducted abroad, while these kinds of studies were seldom conducted in China. Therefore, data from the China National Nutrition and Health Surveillance (CNNHS) in 2010–2012 were utilized to explore the relationship between SB and metabolic risks in middle-aged women.

## 2. Materials and Methods

### 2.1. Study Participants

The data came from the China National Nutrition and Health Surveillance (CNNHS) in 2010–2012. The multi-stage stratification method and the population proportional cluster random sampling method was used. In the first stage, a total of 150 study sites were selected from the four categories of areas, including 34 large cities, 41 small and medium-sized cities, 45 ordinary rural areas, and 30 poor rural areas. In the second stage, six villages or communities were selected from each site by using the proportional method of population. In the third stage, 75 households were randomly selected from each selected village or community.

All family members in each selected family were included as respondents after signing the informed consent form [23]. In the present study, a total of 2643 middle-aged (ages 46 to 53) women with complete blood glucose, blood lipid, blood pressure, height, weight, and dietary information were selected as participants. This study was approved by the ethics review committee of the National Institute for Nutrition and Health, Chinese Center for Disease Control and Prevention (No. 2013-018); all participants signed the informed consent.

### 2.2. Anthropometric Measurements

Height was measured in centimeters with an accuracy of 0.1 cm. Fasting weight was measured in kilograms with an accuracy of 0.1 kg. Waist circumference (WC) was measured twice, and the mean value was taken (the accuracy was 0.1 cm). A standard mercury sphygmomanometer (scale 0 to 300 mmHg) was applied to measure blood pressure (BP). Systolic and diastolic blood pressure (SBP and DBP) were determined by the onset (KrotkoFF phase I) and the disappearance (KrotkoFF phase V) of sound, respectively. BP was measured three times, and the average of the three readings was calculated for the final analysis. Blood samples harvested from veins was checked for fasting blood glucose (FBG), oral glucose tolerance (OGTT), and serum lipids [23].

### 2.3. Categories of LTSB

In the present study, LTSB included watching TV, using a computer, playing video games, reading, and doing homework in spare time, information on which was collected using an interview-administrated questionnaire. The participants were asked to recall the average total daily amount of LTSB over the past year. The values of LTSD at the 25th and 75th percentiles of a frequency distribution divided the data into three parts. LTSD < 2 h was classified as low-level LTSB, 2 h > LTSD < 3 h was classified as middle-level LTSB, and LTSD ≥ 3 h was classified as high-level LTSB.

### 2.4. Definitions of Outcome Variables

#### 2.4.1. Diabetes

China’s national guidelines for the prevention and control of diabetes in primary care (2018) define diabetes as FBG ≥ 7.0 mmol/L or OGTT ≥ 11.1 mmol/L [9].

#### 2.4.2. Dyslipidemia

Total cholesterol (TC) ≥ 6.2 mmol/L was defined as hypercholesterolemia. Triglyceride (TG) ≥ 2.3 mmol/L was defined as hypertriglyceridemia. High-density lipoprotein cholesterol (HDL-C) < 1.0 mmol/L was defined as low HDL-C [10].

#### 2.4.3. Hypertension

According to the Chinese guideline for the prevention and treatment of hypertension, the diagnostic criteria for hypertension were SBP ≥ 140 mmHg and/or DBP ≥ 90 mmHg [24].

#### 2.4.4. Overweight and Obesity

Chinese guidelines for the prevention and control of overweight and obesity for adults classify body mass index (BMI) < 18.5 kg/m^2^ as underweight, BMI between 18.5 kg/m^2^ and 23.9 kg/m^2^ as normal, BMI between 24.0 kg/m^2^ and 27.9 kg/m^2^ as overweight, and BMI ≥ 28 kg/m^2^ as obese. Central obesity was defined as waist circumference ≥80 cm for females [25].

#### 2.4.5. Metabolic Syndrome

The diagnostic criteria for MetS were based on the National Cholesterol Education Program Expert Panel (NCEP) and Adult Treatment Panel III (ATP III), which were modified in 2004 and adapted for Asians. The definition of MetS consists of the clinical condition meeting at least three of the following risk factors: WC ≥ 80 cm, HDL-C < 1.0 mmol/L or under treatment, TG ≥ 1.7 mmol/L or under treatment, increased blood pressure >130/85 mmHg or under treatment, and FBG≥ 5.6 mmol/L or under treatment [26].

### 2.5. Statistical Analyses

Statistical analysis was performed using the Statistical Analysis System (SAS) 9.4 software (SAS Institute Inc., Cary, NC, USA). The continuous variables were described by means and standard deviation and analyzed by linear regression. The categorical variables were described by rate and analyzed by chi-square test. Simple linear regression and multiple linear regression were used to analyze the relationship between LTSD and TC content, TG content, HDL-C content, WC, and BMI. Educational level (primary school and below = 0, junior high school = 1, senior high school and above = 2), occupation (employer = 1, others = 2, farmer = 3), and economic level (per capita annual income <CNY 20,000 was the low economic level, between CNY 20,000 and CNY 40,000 was the middle economic level, >CNY 40,000 was the high economic level), drinking (no drinking = 0, drinking = 1), leisure exercise level (no leisure exercise = 0, leisure exercise = 1), energy intake (kcal), and fat energy ratio (%) were included in the model as confounders.

In previous studies, SB was usually incorporated into the logistic model as a categorical variable; however, when a continuous variable is converted into a categorical variable, some of the original information is often lost, especially in the study of biological clinical indicators [27]. When the variable is continuous, a nonlinear correlation method is strongly recommended, and restrictive cubic splines (RCS) is a good method. This approach was used in some studies to examine the association of certain continuous variables (such as shift years [28] and sleep duration [29]) with disease, as well as the relationship between sedentary behaviors and mortality in patients with type 2 diabetes [30]. Therefore, in the present study, we applied this method to investigate the relationship between LTSD as a continuous variable and the risk of diabetes, MetS, hypercholesteremia, hypertriglyceridemia, low HDL-C level, hyperglycemia, hypertension, central obesity, overweight, and obesity. In addition, a limitation of the RCS method is reflected by using a large number of knots, which may lead to “overfitting” the data [27]. Therefore, we only chose three knots in the analysis to avoid such a phenomenon.

## 3. Results

### 3.1. Baseline Characteristics of the Study Population

A total of 2643 participants were investigated. Overall, 17.5% had a low-level LTSB, 36.4% had a middle-level LTSB, and 46.1% had a high-level LTSB. The level of LTSB varied from urban to rural (*p* < 0.05), and in those with different levels of education (*p* < 0.05). With an increase in LTSB level, TC content, TG content, and the prevalence of hyperlipidemia increased (Table 1).

### 3.2. The Relationship between LTSB and TC, TG, HDL-C, WC, and BMI

Table 2 shows the association of LTSD with TC and TG content, WC, BMI, and HDL-C. The results of simple linear regression showed that, for every hourly increase in LTSD, TC and TG increased by 0.04 mmol/L and 0.03 mmol/L respectively. After adjusting for potential influencing factors, multiple linear regression results showed that, for every hourly increase in LTSD, TC and TG increased by 0.03 mmol/L (*p* < 0.05) and 0.04 mmol/L (*p* < 0.05), respectively.

### 3.3. RCS Curves for LTSD and Metabolic Diseases

Figure 1 shows the RCS curves for LTSD with respect to the risks of metabolic diseases, while the odds ratio (OR) and 95% confidence interval (CI) of statistically significant outcome variables are listed in Table 3, Table 4, Table 5 and Table 6. The risks of MetS (*p* for trend = 0.0276), obesity (*p* for trend = 0.0369), hypertension (*p* for trend = 0.0062), and hypercholesteremia (*p* for trend = 0.0033) gradually increased with each additional unit of increased LTSD.

## 4. Discussion

In this cross-sectional study of middle-aged women, we found that an increase in TC and TG content was associated with the increase in LTSD. The odds ratios of MetS, obesity, hypertension, and hypercholesteremia were constantly on the rise with the increase in LTSD. LTSB is a kind of static behavior that, in addition to having a low energy expenditure, may also underlie a deleterious influence on other lifestyle and eating behaviors. For example, foods are more accessible while sitting at a table or desk than when engaging in other activities such as walking or doing housework, which may lead to an unconscious increase in food intake. At present, watching TV, using computers, and using mobile phones are becoming the main recreational activities for people in their leisure time, which makes people more susceptible to temptation from advertisements for energy-dense foods on their electronic equipment, leading to them being more likely to eat more calories or form unhealthy eating patterns [13,31]. In addition, the amount of time available for other physical activities is compressed for a prolonged sedentary duration, which can contribute to metabolic disease if excess energy cannot be consumed in time. Another reason for the detrimental effect of LTSB on metabolic diseases is that LTSBs cause metabolic dysfunction. SB inhibits lipoprotein lipase (LPL) activity, which is an enzyme that promotes the uptake of free fatty acids into skeletal muscle and adipose tissue [32] Low LPL levels can partially account for the elevated levels of circulating triglycerides, the reduced levels of HDL cholesterol, and the increased risk of cardiovascular disease [32]. Moreover, SB can also affect carbohydrate metabolism by changing muscle glucose transporter (GLUT) protein content, which is essential to glucose uptake [32]. With prolonged sedentary duration, reduced skeletal muscle contraction may lead to decreased lipoprotein lipase activity, decreased triglyceride clearance, decreased oral glucose load clearance, and decreased glucose-stimulated insulin secretion [33,34,35].

The relationship between SB and metabolic diseases was discussed in previous studies; however, the components of SB vary. Most researchers paid more attention to the relationship between watching TV and metabolic diseases [19,36], but other SBs should not be ignored. It seems more suitable to include all SBs rather than particular kinds of SBs when estimating the association of LTSB with metabolic diseases. Especially in this rapidly developing society, smart phones and computers have largely replaced televisions as new forms of entertainment, such that using TV time as a proxy for total LTSD paints only a partial picture. In this study, SB referred to total LTSD. Moreover, we considered LTSD as a continuous variable by using RCS regression to explore its relationship with metabolic risks rather than as a categorical variable by using logistic regression (as done in most previous research), allowing us to preserve the original characteristics of the data. We also adjusted for demographic characteristics, alcohol consumption, exercise, and dietary factors, thereby improving the reliability of the results.

Our study also had some limitations. First, this was a cross-sectional study and, as such, the causal relationship between LTSB and metabolic diseases could not be determined. Second, LTSD was obtained through questionnaires, with limited accuracy. Third, active physical activity might have a positive effect on metabolic diseases. However, we only adjusted for whether the participants exercised regularly in their spare time or not. The quantity and the intensity of physical activity were not determined. Overall, the present study still provides important clues and supplementary evidence for future prospective studies and randomized clinical trials on the relationship between SB and metabolic diseases. It is of great significance for preserving and enhancing the health of middle-aged women.

## 5. Conclusions

The results of the present study suggest that higher LTSB is associated with the presence of MetS, obesity, hypertension, and hypercholesteremia in middle-aged women. Considering that middle-aged women are susceptible to metabolic diseases, this study has important implications for addressing prolonged LTSD as a new public health issue in women at that age.

## Figures and Tables

**Figure 1 ijerph-17-07171-f001:**
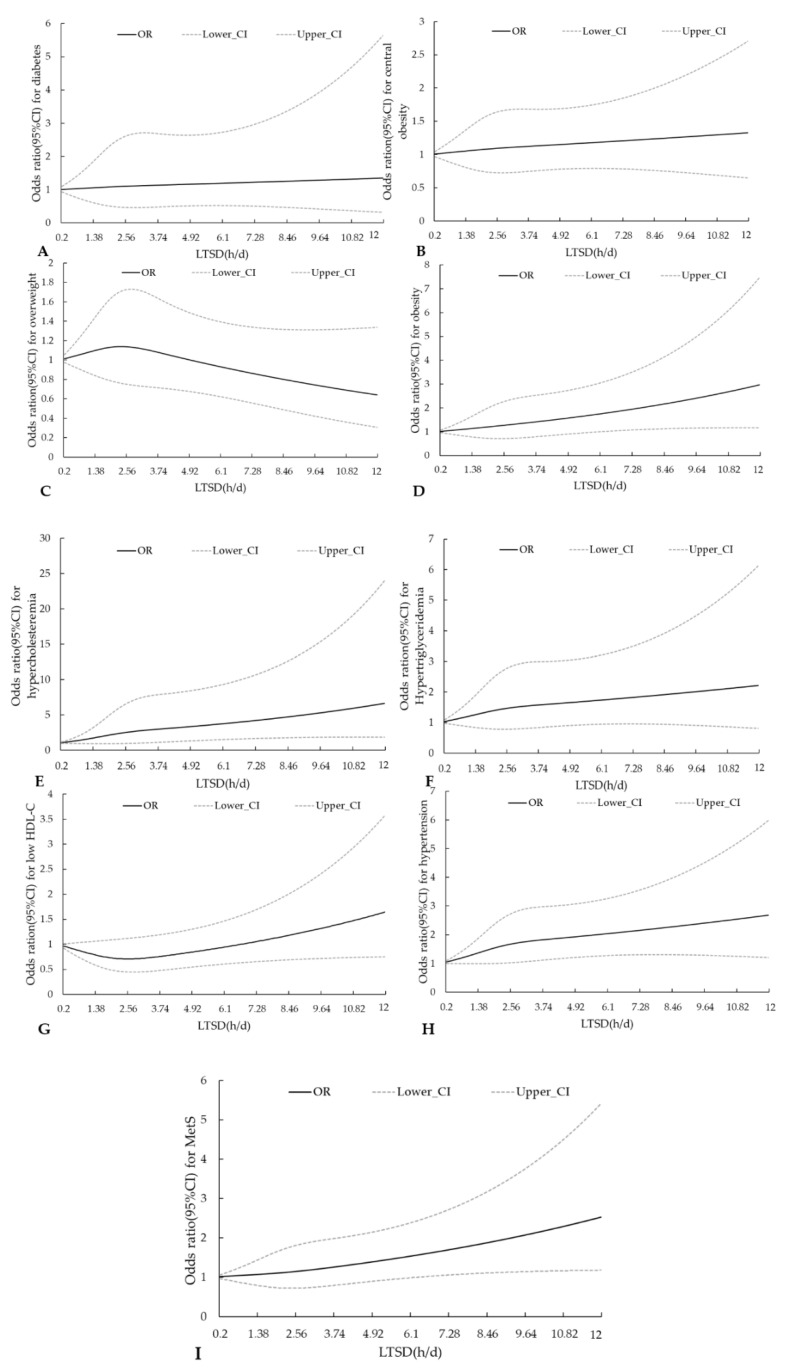
Restrictive cubic spline (RCS) curves for sedentary duration and (**A**) diabetes, (**B**) central obesity, (**C**) overweight, (**D**) obesity, (**E**) hypercholesterolemia, (**F**) hypertriglyceridemia, (**G**) low HDL-C level, (**H**) hypertension, and (**I**) metabolic syndrome (MetS). Region, education, income, alcohol consumption, exercise, daily energy intake, and fat energy ratio were adjusted for all RCS curves.

**Table 1 ijerph-17-07171-t001:** Basic characteristics between middle-aged women with different leisure-time sedentary behavior (LTSB) levels.

Variables	Low-Level LTSB (<2.0 h/day)	Middle-Level LTSB (2.0–3.0 h/day)	High-Level LTSB (≥3.0 h/day)	*p*
Total	463 (17.5%)	962 (36.4%)	1218 (46.1%)	
Residence (%)				
Urban	47.5	45.5	57.2	<0.0001
Rural	52.5	54.5	42.8	
Education (%)				
Primary school and low	46.4	37.9	30.9	<0.0001
Middle school	33.7	36.5	37.7	
High school and above	19.9	25.6	31.4	
Family income				
Low income	50.8	46.9	44.6	0.184
Middle income	36.5	36.0	39.2	
High income	8.0	10.8	10.8	
Unknown	4.8	6.3	5.5	
BMI (kg/m²)	24.3 ± 13.3	24.6 ± 3.3	24.6 ± 3.6	0.097
WC (cm)	81.3 ± 9.0	81.5 ± 8.9	81.3 ± 9.3	0.203
Leisure exercise (%)	13.8	13.6	15.3	0.509
Drinking (%)	19.2	17.3	17.2	0.583
Energy intake (kcal)	1912.8 ± 737.6	1936.2 ± 743.2	1908.7 ± 818.8	0.485
Fat energy ratio (%)	26.6 ± 13.3	27.2 ± 12.8	27.7 ± 12.8	0.000
TC (mmol/L)	4.66 ± 0.93	4.72 ± 0.93	4.83 ± 0.96	0.007
TG (mmol/L)	1.38 ± 0.91	1.43 ± 0.99	1.49 ± 1.07	0.031
HDL-C (mmol/L)	1.21 ± 0.34	1.22 ± 0.32	1.22 ± 0.33	0.264
Hypercholesteremia (%)	4.3	5.2	8.2	0.002
Hypertriglyceridemia (%)	11.0	12.6	13.1	0.528
Low HDL-C level (%)	29.2	25.0	25.9	0.230
Diabetes (%)	5.2	6.8	5.9	0.477
Hypertension (%)	20.7	25.9	25.9	0.068
BMI status				<0.0001
Underweight	2.2	2.4	2.4	
Normal	46.2	40.8	43.3	
Overweight	37.4	42.6	38.1	
Obesity	14.3	14.2	16.3	
Central Obesity (%)	55.3	55.9	55.7	0.975
Metabolic syndrome (%)	25.7	27.4	28.2	0.601

Note: BMI: b*ody* m*ass* i*ndex;* WC: waist circumference; TC: total cholesterol; TG: triglyceride; HDL-C: high-density lipoprotein cholesterol.

**Table 2 ijerph-17-07171-t002:** Linear regression analysis of the relationship between leisure-time sedentary duration (LTSD) and TC, TG, HDL-C, WC, and BMI.

Variables	Simple Linear Regression		Multiple Linear Regression ^1^	
	β_1_	*p* _1_	β_2_	*p* _2_
TC	0.04	0.004	0.03	0.019
TG	0.03	0.030	0.04	0.015
HDL-C	0.00	0.534	0.00	0.336
WC	0.15	0.253	0.23	0.076
BMI	0.08	0.112	0.10	0.055

Note: **β_1:_** simple linear regression coefficient; ***p*****_1_**: p value of simple linear regression; **β_2:_** multiple linear regression coefficient. ***p*****_2_**: p value of multiple linear regression. ^1^ Region, education, income, alcohol consumption, exercise, daily energy intake, and fat energy ratio were adjusted for all multiple linear regression models.

**Table 3 ijerph-17-07171-t003:** Odds ratio (OR) and 95% confidence interval (CI) for the restrictive cubic spline of MetS.

LTSD	OR	95% CI	*p* for Trend
1.3	1.06	(0.79–1.42)	0.0276
2.3	1.12	(0.72–1.75)	
3.3	1.21	(0.76–1.93)	
4.3	1.32	(0.85–2.06)	
5.3	1.44	(0.93–2.22)	
6.3	1.56	(1.00–2.44)	
7.3	1.70	(1.06–2.72)	
8.3	1.85	(1.10–3.10)	
9.3	2.01	(1.13–3.58)	
10.3	2.19	(1.15–4.16)	
11.3	2.39	(1.17–4.87)	

**Table 4 ijerph-17-07171-t004:** OR and 95% CI for the restrictive cubic spline of obesity.

LTSD	OR	95% CI	*p* for Trend
1.0	1.10	(0.83–1.47)	0.0369
2.0	1.21	(0.72–2.05)	
3.0	1.33	(0.73–2.42)	
4.0	1.46	(0.82–2.58)	
5.0	1.59	(0.91–2.76)	
6.0	1.75	(1.00–3.04)	
7.0	1.90	(1.07–3.39)	
8.0	2.08	(1.11–3.88)	
9.0	2.27	(1.15–4.51)	
10.0	2.49	(1.17–5.31)	
11.0	2.72	(1.17–6.31)	

**Table 5 ijerph-17-07171-t005:** OR and 95% CI for the restrictive cubic spline of hypertension.

LTSD	OR	95% CI	*p* for Trend
2.0	1.54	(1.00–2.38)	0.0062
3.0	1.75	(1.06–2.89)	
4.0	1.85	(1.14–2.99)	
5.0	1.94	(1.22–3.08)	
6.0	2.03	(1.27–3.24)	
7.0	2.13	(1.30–3.48)	
8.0	2.23	(1.31–3.80)	
9.0	2.34	(1.30–4.21)	
10.0	2.45	(1.28–4.71)	
11.0	2.57	(1.24–5.31)	

**Table 6 ijerph-17-07171-t006:** OR and 95% CI for the restrictive cubic spline of hypercholesteremia.

LTSD	OR	95% CI	*p* for Trend
1.7	2.02	(0.94–4.34)	0.0033
2.7	2.64	(1.00–6.96)	
3.7	3.00	(1.14–7.88)	
4.7	3.31	(1.31–8.32)	
5.7	3.64	(1.47–8.99)	
6.7	4.01	(1.62–9.97)	
7.7	4.42	(1.73–11.33)	
8.7	4.87	(1.80–13.17)	
9.7	5.37	(1.85–15.60)	
10.7	5.92	(1.87–18.77)

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
