# Peer review of "The Relationship between Leisure-Time Sedentary Behaviors and Metabolic Risks in Middle-Aged Chinese Women"

_ijerph, 2020, doi:10.3390/ijerph17197171_

Round 1
Reviewer 1 Report
This submission investigated the relationship between leisure-time sedentary behaviors (LTSB) and metabolic risks in middle-aged women in China. I like to give the following comments.
- The leisure-time sedentary behaviors (LTSB) did not introduce. Why?
- How to link LTSB with metabolic syndrome? Please introduce in clear.
- Why focused on the middle-aged women? It needs a rationale.
- Daily LTSB method must follow the previous report(s).
- RCS curves used in analysis also need the reference(s) to support. Additionally, limitation of this application was not discussed.
- LTSB may also be a risk factor for metabolic diseases among another aged people. How did you conclude it for middle-aged women only?
- LTSB caused metabolic dysfunction that needs more reports to support.
Reviewer 2 Report
ABSTRACT
- (and 42 and …) Remove the space after the comma when typing numbers in the thousands.
27-33. The last two sentences are misleading and dependent on sample size. See lines 167-172 below.
INTRODUCTION
- Replace “leaded” with “led”
62-63. “The current evidence from many epidemiological studies [were] is consistent with the [results] hypothesis that SB is increasing the risk of type2 diabetes[11-13]and obesity[14-17].”
- Replace “undertaken” with “undertook”
MATERIALS AND METHODS
- Maintain past tense throughout. Replace “are” with “were”
104-105. “Daily LTSB (watching TV, using computer, playing video games, reading, etc.) was collected by interview-administrated questionnaire.”
Please provide more detail. How was the questionnaire phrased? Did you query each sedentary activity separately? Did you ask about activity “on an average day” or “over the past 7 days”, etc? It would be useful to include your actual questionnaire as a supplement, or to include it in the main text if it is simple.
- Replace “referred to” with “defined as”
- Leisure exercise level defined as 1 or 0 is a very limited measure of physical activity. A better measure would be the PASE score [Washburn et al. 1993. THE PHYSICAL ACTIVITY SCALE FOR THE ELDERLY (PASE): DEVELOPMENT AND EVALUATION, J Clin Epidemiol Vol. 46, No. 2, pp. 153-l 62], or at the least, some measure of duration and intensity of physical activity. (Too late now, but should definitely be mentioned in the limitations.)
RESULTS
167-172. “When the daily LTSD approximately equaled to 6.3 hours, 6 hours, 2 hours and 2.7 hours, respectively, the risk of MetS(OR=1.56, 95%CI:1.00-2.44), obesity(OR=1.75, 95%CI:1.00-3.04), hypertension(OR=1.54, 95%CI:1.00-2.38) and hypercholesteremia(OR=2.64,95%CI:1.00-6.96) began to increase. Subsequently, the risk of MetS, obesity, hypertension and hypercholesteremia gradually increased with increased LTSD, and the difference was statistically significant.”
This seems to say that the Odds Ratio remained at 1.0 for each of these conditions until LTSD levels of 6.3, 6.0, 2.0 and 2.7 hours, respectively were reached, and only then did the OR begin to increase. However, that is not the story we see in either your RCS curves or in tables 3-6. They, instead, reveal a remarkably monotonic increase in the odds with each increase in LTSD at any level. Your values of 6.3, 6.0 hours, etc. are merely telling us the point at which, in your particular data with your particular sample size, the lower bound of the confidence interval reached 1.0. If you had double the sample size, all else equal, you would have reached “significance” at a lower level of LTSD.
It would be misleading to say, as you seem to, that LTSD of 5 hours carries no risk of MetS relative to 1 hour.
You have some very nice data here. Rather than focusing on the exact point at which this nice curve crosses OR=1.0, I think you would do better to show your curves and tables as they are. Just remove the bold typeface in the tables for the values where OR begins to be “significantly” greater than 1. Also, report the overall “p for trend” (or the overall p-value from the logistic regression of condition on LTSD, if linearity assumptions are not violated, as seems to be the case for nearly all of your variables). Now your results are more generalizable and not dependent on sample size. Your true findings were that the odds of these conditions increased with each additional unit of LTSD.
DISCUSSION
190-194. “After a certain length of LTSD, the relationship between LTSD and the risk of metabolic diseases began to be statistically significant, and with the continuous increase of LTSD, the risk of metabolic diseases increased. Thus, the results of our study suggested that only when LTSB reached a certain level did it have a negative effect on metabolic diseases.”
No, see above. You did not demonstrate a threshold level for LTSD in any of these conditions. If you had, your curve would be shaped like this:
___/
220-223. “This approach gave the exact time point at which the relationship between LTSD and metabolic disease became statistically significant, which provides important clues to the determination of SD classification points for further study and the formulation of relevant policies in the future.”
No, see above.
- Add to limitation discussion. You included only a yes/no indicator of leisure-time exercise as a covariate in the multiple regressions. This is better than nothing but very limited. It is conceivable that most of your effect of LTSB on health conditions is actually due to the duration and intensity of the physical activity that people engaged in during their non-sedentary periods. We cannot conclude from this study that it was the sedentary behavior per se that put people at risk.
CONCLUSION
232-234. The increase in metabolic risks was statistically significant only if the duration of LTSB was long enough, and the amount of LTSD it takes for different metabolic diseases to develop is different.
Error in interpretation, see above.
Author Response
Response to Reviewer 2 Comments
Thank you very much for your valuable suggestions and comments. The detailed response for each point please see the bellows.
Point 1: ABSTRACT
(and 42 and …) Remove the space after the comma when typing numbers in the thousands.
27-33. The last two sentences are misleading and dependent on sample size. See lines 167-172 below.
INTRODUCTION
Replace “leaded” with “led”
62-63. “The current evidence from many epidemiological studies [were] is consistent with the [results] hypothesis that SB is increasing the risk of type2 diabetes[11-13]and obesity[14-17].”
Replace “undertaken” with “undertook”
MATERIALS AND METHODS
Maintain past tense throughout. Replace “are” with “were”
Replace “referred to” with “defined as”
Response 1: We thanks for your careful reviewing and corrected all the grammar mistakes you marked.
Point 2:104-105. “Daily LTSB (watching TV, using computer, playing video games, reading, etc.) was collected by interview-administrated questionnaire.”
Please provide more detail. How was the questionnaire phrased? Did you query each sedentary activity separately? Did you ask about activity “on an average day” or “over the past 7 days”, etc? It would be useful to include your actual questionnaire as a supplement, or to include it in the main text if it is simple.
Response 2: The LTSB phrased in our questionnaire refers to all the sedentary behaviors in leisure time, such as reading, using the computer, watching TV and doing homework, etc. We did not query each sedentary activity separately. We asked about the total amount of time they engaged in those behaviors on an average day over the past year. We gave a complete definition of LTSB in detail in the revised manuscript as “In the present study, LTSB included watching TV, using a computer, playing video games, reading, and doing homework and so on in spare time, which was collected by interview-administrated questionnaire. The participants were asked to recall the average daily total amount of LTSB over the past year.” (line 130-133)
Point 3:Leisure exercise level defined as 1 or 0 is a very limited measure of physical activity. A better measure would be the PASE score [Washburn et al. 1993. THE PHYSICAL ACTIVITY SCALE FOR THE ELDERLY (PASE): DEVELOPMENT AND EVALUATION, J Clin Epidemiol Vol. 46, No. 2, pp. 153-l 62], or at the least, some measure of duration and intensity of physical activity. (Too late now,but should definitely be mentioned in the limitations.)
Response 3: We added this limitation in the discussion section in the revised manuscript as “Third, active physical activity might have a positive effect on metabolic diseases. But we only adjusted for whether the participants did some regular physical exercise in their spare time or not. The quantity and intensity of physical activity was lack.”(Line 284-287).
Point 4:167-172. “When the daily LTSD approximately equal to 6.3 hours, 6 hours, 2 hours and 2.7 hours, respectively, the risk of MetS(OR=1.56, 95%CI:1.00-2.44), obesity(OR=1.75, 95%CI:1.00-3.04), hypertension(OR=1.54, 95%CI:1.00-2.38) and hypercholesteremia(OR=2.64,95%CI:1.00-6.96) began to increase. Subsequently, the risk of MetS, obesity, hypertension and hypercholesteremia gradually increased with increased LTSD, and the difference was statistically significant.”
This seems to say that the Odds Ratio remained at 1.0 for each of these conditions until LTSD levels of 6.3, 6.0, 2.0 and 2.7 hours, respectively were reached, and only then did the OR begin to increase. However, that is not the story we see in either your RCS curves or in tables 3-6. They, instead, reveal a remarkably monotonic increase in the odds with each increase in LTSD at any level. Your values of 6.3, 6.0 hours, etc. are merely telling us the point at which, in your particular data with your particular sample size, the lower bound of the confidence interval reached 1.0. If you had double the sample size, all else equal, you would have reached “significance” at a lower level of LTSD.
It would be misleading to say, as you seem to, that LTSD of 5 hours carries no risk of MetS relative to 1 hour.
You have some very nice data here. Rather than focusing on the exact point at which this nice curve crosses OR=1.0, I think you would do better to show your curves and tables as they are. Just remove the bold typeface in the tables for the values where OR begins to be “significantly” greater than 1. Also, report the overall “p for trend” (or the overall p-value from the logistic regression of condition on LTSD, if linearity assumptions are not violated, as seems to be the case for nearly all of your variables). Now your results are more generalizable and not dependent on sample size. Your true findings were that the odds of these conditions increased with each additional unit of LTSD.
190-194. “After a certain length of LTSD, the relationship between LTSD and the risk of metabolic diseases began to be statistically significant, and with the continuous increase of LTSD, the risk of metabolic diseases increased. Thus, the results of our study suggested that only when LTSB reached a certain level did it have a negative effect on metabolic diseases.”
No, see above. You did not demonstrate a threshold level for LTSD in any of these conditions. If you had, your curve would be shaped like this:
___/
220-223. “This approach gave the exact time point at which the relationship between LTSD and metabolic disease became statistically significant, which provides important clues to the determination of SD classification points for further study and the formulation of relevant policies in the future.”
No, see above.
The increase in metabolic risks was statistically significant only if the duration of LTSB was long enough, and the amount of LTSD it takes for different metabolic diseases to develop is different.
Error in interpretation, see above.
Response 4: Thank you very much for your valuable comments. We removed the bold typeface in the tables for the values where OR begins to be “significantly”. We reported the overall “p for trend” in the revised manuscript and removed all the misleading statements and redescribed our results in the different parts of this article. Here are the changes we made:
- Abstract:
"The results of RCS curves showed that the risk of MetS(OR=1.56, 95%CI:1.00-2.44), obesity(OR=1.75, 95%CI:1.00-3.04), hypertension(OR=1.54,95%CI:1.00-2.38) and hypercholesteremia(OR=2.64, 95%CI:1.00-6.96) increased with the increase of LTSD when it was approximately up to 6.3 hours, 6 hours, 2 hours and 2.7 hours, respectively. LTSB is a risk factor for metabolic disease in middle aged women and it may not have a negative effect on metabolic disease until a certain level of LTSD was reached." has been changed into "The results of RCS curves showed that the risks of MetS(p for trend=0.0276), obesity(p for trend=0.0369), hypertension(p for trend=0.0062) and hypercholesteremia (p for trend=0.0033) increased with the increase of LTSD".(line 30-37)
- Result
"Figure 1 showed RCS curves between LTSD and the risks of metabolic diseases, and the OR value and 95%CI of statistically significant outcome variables were listed in Table 3-6.When the daily LTSD approximately equaled to 6.3 hours, 6 hours, 2 hours and 2.7 hours, respectively, the risk of MetS (OR=1.56, 95%CI:1.00-2.44), obesity(OR=1.75, 95%CI:1.00-3.04), hypertension (OR=1.54, 95%CI:1.00-2.38) and hypercholesteremia (OR=2.64, 95%CI:1.00-6.96) began to increase. Subsequently, the risk of MetS, obesity, hypertension and hypercholesteremia gradually increased with each additional unit of LTSD, and the difference was statistically significant." has been changed into "Figure 1 showed RCS curves between LTSD and the risks of metabolic diseases, and the OR value and 95%CI of statistically significant outcome variables were listed in Table 3-6.The risks of MetS (p for trend=0.0276)), obesity (p for trend=0.0369), hypertension (p for trend=0.0062) and hypercholesteremia (p for trend=0.0033) gradually increased with each additional unit of increased LTSD." (line 211-222)
In addition, we added a new column (tag: p for trend) to tables 3-6 and reported the values.
- Discussion
"After a certain length of LTSD, the relationship between LTSD and the risk of metabolic diseases began to be statistically significant, and with the continuous increase of LTSD, the risk of metabolic diseases increased. Thus, the results of our study suggested that only when LTSB reached a certain level did it have a negative effect on metabolic diseases. (line 288-231)" has been changed into "The odds ratio of MetS, obesity, hypertension and hypercholesteremia were constantly on the rise with the increase of LTSD."(line 241-246)
"This approach gave the exact time point at which the relationship between LTSD and metabolic disease became statistically significant, which provides important clues to the determination of SD classification points for further study and the formulation of relevant policies in the future. (line 277-280)" was deleted.
- Conclusion:
"LTSB was a risk factor for metabolic disease among Chinese middle-aged women. The increase in metabolic risks was statistically significant only if the duration of LTSB was long enough, and the amount of LTSD it takes for different metabolic diseases is different. Considering middle-aged women are susceptible to metabolic diseases, it has important implications for addressing prolonged LTSD as a new public health issue in women at that age.(line 296-302)" has been changed into "The results of the present study suggest that higher LTSB is associated with the presence of MetS, obesity, hypertension and hypercholesteremia in middle-aged women. Considering middle-aged women are susceptible to metabolic diseases, it has important implications for addressing prolonged LTSD as a new public health issue in women at that age.". (line 293-299)
Point 6: Add to limitation discussion. You included only a yes/no indicator of leisure-time exercise as a covariate in the multiple regressions. This is better than nothing but very limited. It is conceivable that most of your effect of LTSB on health conditions is actually due to the duration and intensity of the physical activity that people engaged in during their non-sedentary periods. We cannot conclude from this study that it was the sedentary behavior per se that put people at risk.
Response 6: We added this limitation in the discussion section in the revised manuscript as “Third, active physical activity might have a positive effect on metabolic diseases. But we only adjusted for whether the participants did some regular physical exercise in their spare time or not. The quantity and intensity of physical activity was lack”. (Line 284-287).
However, some previous studies showed that SB related to metabolic diseases, independent of physical activity[Bennett, D.A.; Du, H.; Bragg, F.; Guo, Y.; Wright, N.; Yang, L.; Bian, Z.; Chen, Y.; Yu, C.; Wang, S., et al. Physical activity, sedentary leisure-time and risk of incident type 2 diabetes: a prospective study of 512 000 Chinese adults. BMJ Open Diabetes Res Care 2019, 7, e000835, doi:10.1136/bmjdrc-2019-000835.] [Bankoski A, Harris TB, McClain JJ, et al. Sedentary activity associated with metabolic syndrome independent of physical activity. Diabetes Care. 2011;34(2):497-503. doi:10.2337/dc10-0987] [Owen N, Sparling PB, Healy GN, Dunstan DW, Matthews CE. Sedentary behavior: emerging evidence for a new health risk. Mayo Clin Proc. 2010;85(12):1138-1141. doi:10.4065/mcp.2010.0444.]. Thus, we may conclude in the present study that higher LTSB is associated with the presence of MetS, obesity, hypertension and hypercholesteremia in middle-aged women.
Round 2
Reviewer 2 Report
Thank you for addressing all of my previous concerns. I am just puzzled by one section in your methods concerning the use of restrictive cubic splines and three knots. I don't see any indication that you actually used this to obtain your results. They look like the result of simple linear regression (which is fine, because the ORs appear to increase linearly). Am I missing something? Or did you not use RCS? If you didn't use it, please omit this paragraph.
Author Response
Response to Reviewer 2 Comments
Point 1: Thank you for addressing all of my previous concerns. I am just puzzled by one section in your methods concerning the use of restrictive cubic splines and three knots. I don't see any indication that you actually used this to obtain your results. They look like the result of simple linear regression (which is fine, because the ORs appear to increase linearly). Am I missing something? Or did you not use RCS? If you didn't use it, please omit this paragraph.
Response 1: Thank you very much for checking our response and giving further comments. The real relationship between LTSD and metabolic diseases was not very clear so we wanted to explore it by using RCS which can test both linearity and nonlinearity. When using RCS to draw the curve, it is usually necessary to set the number and the position of spline function(knots). In most cases, the position of knots has little influence on the fitting of restrictive cubic spline, while the number of knots determines the shape or smoothness of the curve. Most researchers recommend 3-7 knots (Lou J.F., et al. The Application of Restricted Cubic Spline in Nonlinear Regression. Chinese Journal of Health Statistics2010. Doi: 10.3969/j.issn.1002-3674.2010.03.002). Thus, we used RCS with three knots (the 10th, 50th and 90th percentile) to flexibly model and visualize the relation of LTSD and metabolic diseases.
We re-ran the SAS program to do the nonlinear test. Here are the specific results:
Table 1. The contrast results of RCS for MetS.
|
Contrast Results |
||||
|
Contrast |
DF |
Chi-Square |
Pr > ChiSq |
Type |
|
Overall association |
2 |
5.75 |
0.0564 |
Wald |
|
Non-linear association |
1 |
0.07 |
0.7843 |
Wald |
Table 2. The contrast results of RCS for Obesity.
|
Contrast Results |
||||
|
Contrast |
DF |
Chi-Square |
Pr > ChiSq |
Type |
|
Overall association |
2 |
5.57 |
0.0618 |
Wald |
|
Non-linear association |
1 |
0.00 |
0.9707 |
Wald |
Table 3. The contrast results of RCS for Hypertension.
|
Contrast Results |
||||
|
Contrast |
DF |
Chi-Square |
Pr > ChiSq |
Type |
|
Overall association |
2 |
9.06 |
0.0108 |
Wald |
|
Non-linear association |
1 |
1.43 |
0.2316 |
Wald |
Table 4. The contrast results of RCS for Hypercholesteremia.
|
Contrast Results |
||||
|
Contrast |
DF |
Chi-Square |
Pr > ChiSq |
Type |
|
Overall association |
2 |
9.76 |
0.0076 |
Wald |
|
Non-linear association |
1 |
1.30 |
0.2548 |
Wald |
RCS is a modelling strategy that imposing the assumption of a non-linear association on a continuous variable. As we can see from table 1-4 here, the relationships between LTSD and the four diseases were linear, but not completely linear, and a little nonlinear but not statistically significant. Overall, we did use RCS method because we thought this was a better way to show the true relationship between LTSD and metabolic diseases.